# Farmers Views on the Implementation of On-Farm Emergency Slaughter for the Management of Acutely Injured Cattle in Ireland

**DOI:** 10.3390/ani13030450

**Published:** 2023-01-28

**Authors:** Paul McDermott, Aideen McKevitt, Flavia H. Santos, Alison J. Hanlon

**Affiliations:** 1Department of Environment, Climate Change and Agriculture, Mayo County Council, Castlebar, F23 WF90 Castlebar, Co. Mayo, Ireland; 2School of Veterinary Medicine, University College Dublin, D04 W6F6 Belfield, Dublin, Ireland; 3School of Agriculture and Food Science, University College Dublin, D04 N2E5 Belfield, Dublin, Ireland; 4School of Psychology, University College Dublin, D04 F6X4 Belfield, Dublin, Ireland

**Keywords:** on-farm emergency slaughter (OFES), casualty slaughter (CS), fitness to transport, farmers, farm animal welfare, perceptions, private veterinary practitioner (PVP)

## Abstract

**Simple Summary:**

There are four options for the management of cattle that experience a severe or acute injury on-farm: treatment, unless cattle are severely injured; on-farm emergency slaughter (OFES); casualty slaughter (CS), where the animal is transported to an abattoir, provided that a vet has certified it fit for transport; or euthanasia. OFES is designed to avoid severely injured animals being transported to the abattoir. An online survey was designed to determine the perceptions of beef and dairy farmers about the management of acutely injured cattle in Ireland. The survey included questions about the farmer (e.g., age, gender) and farm (e.g., herd size), number of acutely injured cattle in 2020, and the frequency of the four management options that were used. Questions on the farmers’ knowledge and experience of, and the cost implications of OFES were included. Other questions related to recommendations to improve OFES. The results were based on responses from 94 farmers: 49 dairy and 45 beef farmers. Not all farmers answered all questions. Most farmers had a positive view of OFES. Negative views related to the poor availability of OFES and that it had a higher overall cost than CS. Increasing availability of OFES and a decreasing cost would help to reduce the number of acutely injured cattle that go for CS.

**Abstract:**

Four management options for acutely injured cattle in Ireland exist: treatment, unless cattle are severely injured; on-farm emergency slaughter (OFES); casualty slaughter (CS) if the animal is certified fit for transport; or euthanasia. OFES is designed to prevent transport of welfare-compromised cattle. An online survey of farmers in Ireland was carried out between April and July 2021 and focused on events during 2020. A theoretical framework of capacity, willingness, and opportunity was used to explore farmers’ perceptions. Responses from 94 farmers (49 dairy and 45 beef) were analysed; not all respondents answered all questions. Respondents indicated that the incidence of acutely injured cattle in Ireland is low. A majority reported not having an acutely injured animal for greater than 36 months. Most respondents had a positive attitude towards OFES for animal welfare reasons and were aware of relevant regulations and guidelines. Barriers to OFES included a lack of availability of OFES, and dairy farmers indicated that it had a similar financial impact as euthanasia. A parallel study with veterinarians indicated a higher incidence of acutely injured cattle in Ireland; the current results may be due to the demographic or the sensitivity of the topic. Nationwide electronic data capture on the cause of mortality could support improvements in the management of acutely injured cattle and enable surveillance of the proportion of these cattle undergoing OFES, euthanasia, or CS.

## 1. Introduction

On-farm emergency slaughter (OFES) “is the slaughter outside the slaughterhouse, of an otherwise healthy animal, which has suffered an accident that, for welfare reasons, prevented its transport to a slaughterhouse” [1]. In Ireland, for OFES to occur, abattoir owners must allow carcasses from acutely injured cattle that have been slaughtered on-farm into their abattoirs for processing. Provided that cattle are deemed suitable for OFES, based on an ante-mortem examination by a private veterinary practitioner (PVP), slaughter is performed, typically on-farm, by either a PVP or a competent slaughter person as required by Regulation (EC) No 1099/2009 [2]. All parts of the carcass must be transported to the abattoir where a post-mortem examination is performed by an official veterinarian. If the meat from an OFES animal is passed as fit for human consumption, it may be placed on the market. The procedure is designed to prevent the transport of acutely injured cattle to abattoirs for casualty slaughter (CS) and provides an alternative to the euthanasia and disposal of injured cattle that are otherwise fit for human consumption [3]. There are both animal welfare and economic implications for farmers who use the OFES [4]. On the other hand, CS is “the slaughter at an abattoir of injured cattle that have been deemed fit for human consumption and fit for transport under veterinary certification” [5]. The management of acutely injured cattle by using OFES is well established in certain parts of Ireland [6]. Legal provision was introduced in Ireland in 2009 to provide an outlet for acutely injured livestock and reduce the prevalence of CS in cattle that were deemed unfit for transport [7]. The procedure requires the agreement of key stakeholders, namely farmers, abattoir owners, official veterinarians, and PVPs [5].

The Structure of Farming in Ireland survey conducted by the Central Statistics Office in 2016 reported that there were 137,500 farms in the Republic of Ireland, of which 111,300 were cattle farms, with a total of 6.2 million cattle, and an average herd size of 62. More than 60% of cattle were located on 52,700 farms in the Southeast region, where the average herd size was 81 cattle. The remaining 38.0% of cattle were located in the Border, Midland, and Western regions and had an average herd size of 45 cattle [8]. There was a total of 228,527 on-farm cattle deaths in the Republic of Ireland in 2020, which is 3.5% of the total population [9]. There is no policy of recording the cause of death of cattle in Ireland.

Throughout Ireland, beef cattle are the most common herd type, accounting for 48% of all herds in 2019 [10]. In contrast, dairy represents 11% of farms, but these herds tend to be larger on average and account for 28.9% of the national herd [10]. A smaller portion of farms (6.1%) were classified as mixed herds, and the store/rearing herd group accounted for 15% of the total herd population. In store herds, animals are bought as youngstock and reared before being sold, primarily to fattening herds for finishing. These fattening herds were the third most common herd category in Ireland at 16.1% [10].

The individual circumstances that affect whether injured cattle are fit to be transported to abattoirs for CS, or whether other options, such as treatment, OFES, or euthanasia, are preferable to reduce the risk of suffering [6], is not just an issue in Ireland [11] but also in other parts of the European Union (EU) [12], the European Economic Area [4], and Canada [13]. The EU, some countries in the European Economic Area, and a number of provinces/territories in Canada have developed regulations and procedures to provide for the OFES of cattle. The European Commission, Directorate General for Health and Food Safety, carried out audits during 2020 and 2021 in a number of EU member states; their main objective was to evaluate the operation of official controls and the enforcement of applicable EU requirements. These included aspects of animal welfare, especially the evaluation of fitness for transport and slaughter. A number of non-compliances were found. In the Netherlands, the audit team was shown examples of fines, starting at 1500 euros, which were imposed for the transport of unfit animals to abattoirs [14]. In Spain, auditors reported evidence of official veterinarians who identified cows in the lairage, considered unfit to travel, but, contrary to regulations, had been transported to the abattoir [15].

The basis of the regulation of OFES in the Nordic countries of Denmark, Finland, Iceland, Norway, and Sweden originates largely from EU legislation. Denmark, Finland, and Sweden are members of the EU, whilst Iceland and Norway are members of the European Economic Area. However, data indicate that the availability and practice of OFES in these countries differs considerably. For example, in Norway, 4.2% of all cattle slaughter is by OFES, whilst in Iceland, OFES has never been recorded [4]. A Norwegian study found that 46% of beef heifers and 17% of dairy heifers were OFES for obstetrical reasons and thus, were not always acutely injured [16]. Domestic slaughter, which is slaughter for home consumption and distinct from OFES, which can be used for commercial purposes, is prevalent in Iceland at 16.2% of all slaughter. Whilst the figures for OFES in Denmark, Finland, and Sweden are unknown, domestic slaughter ranges from 0.9% to 2% [4].

In Canada, it has been reported that the transportation of unfit cattle is a frequent cause of non-compliance with the Health of Animals Regulations [17,18]. OFES typically occurs in southwestern British Columbia, where many dairy farms are located, and it is used for end-of-career dairy cows. A study published in March 2022 by the European Commission found that similar issues occur in the EU in relation to the transport of unfit end-of-career dairy cows (‘cull cows’) [19]. However, OFES may not be performed on cull cows in the EU, as the conditions they are suffering from cannot be considered acute injuries and thus, would be in contravention of EU regulations [1]. The report indicated that illegal transportation does take place; however, the research was unable to identify the scale of the problem [19]. Economic factors were identified as a major driver; for example, it is more expensive to slaughter unfit cows on-farm rather than at abattoirs. There is a financial gain to the farmer in performing CS and selling the carcass. A lack of understanding and varying interpretations of what stakeholders perceive as what constitutes unfit also contributes to unsuitable cows being transported to abattoirs [19]. The welfare of farm animals is of increasing concern to meat producers, as well as consumers and the broader community in Europe, North America, and other parts of the world; however, these groups conceptualize animal welfare in different ways. An Australian study found that producers linked “good welfare” with productivity and profitability and were willing to adopt new practices to improve animal welfare [20]. In another study, Irish stakeholders reported a conflict between a veterinarian’s professional duty to protect animal welfare and their client’s desire to salvage the financial value of cattle through OFES [21].

A recent survey identified that veterinary professionals in Ireland, namely official veterinarians and PVPs, have differing views as to whether OFES provides prompt relief for animal welfare [6]. However, there has been limited research on how farmers perceive OFES. Knowledge and cost are commonly reported as factors influencing farmers’ perceptions of animal welfare [22]. The aim of this research was to explore farmers’ attitudes of the management of acutely injured cattle. We adopted a framework from Coleman and Hemsworth [23] to investigate the influence of capacity, willingness, and opportunity of farmers to use OFES as a method to manage acutely injured cattle. Capacity includes variables such as knowledge, skills, health, and ability. Willingness considers motivation, job satisfaction, attitude towards the animals, and work attitude. Opportunity focuses on working conditions, actions of co-workers, and organisational policies and rules [23].

For the purposes of this and a related study on veterinary professionals [6], an acute injury was defined as an injury that is severe, causes acute pain [24], has a sudden onset, is usually associated with a traumatic event, and is more commonly locomotory.

## 2. Materials and Methods

An online survey was designed to explore the attitudes of farmers in Ireland towards the management of acutely injured cattle. Similar surveys were designed for official veterinarians and PVPs [6].

### 2.1. Materials

#### Design

Survey design followed a process of initial design, refinement, piloting, and further refinement. Quantitative and qualitative questions were developed relevant to capacity, willingness, and opportunity in the context of the management of acutely injured cattle and the provision of OFES in Ireland. Quantitative questions included the demographic profile of participants, such as age, gender, and the number of cattle they own. Qualitative questions were designed to investigate factors influencing farmers’ understanding and experience of OFES. The qualitative questions also consisted of descriptive questions on an 11-point Likert scale and included statements (0 = completely disagree; 10 = completely agree) and ranking questions (ranking of items on importance). Both open-ended and closed questions were used in the surveys. Open-ended questions were used to assess farmer attitudes, such as what they consider to be the positive and negative aspects of OFES in relation to animal welfare as a procedure for dealing with acutely injured cattle. Closed questions offered respondents a number of defined response choices and were used in relation to what informs farmer knowledge about the management of acutely injured cattle. The questionnaire was divided into four parts: demographics of farmers (e.g., gender, age, farm type [dairy, beef, other], farm location); capacity (e.g., education and training, participation in Knowledge Transfer Programmes); willingness (e.g., experience of OFES, including positive and negative attitudes towards OFES); opportunities (e.g., availability of OFES and recommendations to improve uptake of OFES).

Surveys were designed and were followed by an online piloting phase. Four farmers, two beef farmers, one dairy farmer, and one mixed beef/dairy farmer, were asked to complete the survey and a feedback checklist. The only comment received was that some farmers may find it difficult to answer all the questions.

### 2.2. Methods

#### 2.2.1. Recruitment

The survey was distributed by 1. Agriland, Ireland’s largest online farming news portal; 2. The Irish Farmers Journal, Ireland’s largest farming and rural living newspaper, on their social media; 3. The Livestock, Sheep, and Animal Health Committees of the Irish Farmers Association, Ireland’s largest farming representative organisation. An email with links to the survey was sent to these organisations in April 2021, and a reminder was sent in May 2021. In addition, the survey was distributed to students in the School of Agriculture and Food Science, University College Dublin, through the Agriculture Society Committee.

The survey remained open for a ten-week period between April and July 2021. In total, 107 farmers responded.

#### 2.2.2. Data Analysis

Online survey data were automatically uploaded into Microsoft Excel, Version 1908 from Qualtrics^XM^. The data were cleaned, and a codebook of responses created. Each response was given a variable name and a numerical code.

For open-ended questions, responses were scanned for common themes by using thematic analysis [25]. These common themes were also numerically coded.

Data were uploaded into IBM SPSS v27.0^TM^, followed by statistical analysis. Before data analysis could begin, variables were defined.

The relationship between respondent demographics of age, gender, herd size, main farming activity, geographical location, highest level of qualification, and participation in Knowledge Transfer Programmes was investigated using linear logistic regression. Beta (β), the standardized coefficients, were also calculated. These coefficients measure the strength of the effect of each individual independent variable to the dependent variable and range from 0 to 1 or 0 to −1, depending on the direction of the relationship. The relationship between respondents’ knowledge about the procedure of OFES and age, gender, qualification, and post graduate qualifications was investigated using Spearman’s rank order correlation. Spearman’s rho correlation, *r_s_*, can take a range of values from +1 to −1 [26,27]. Explanatory variables were explored using Cramer’s V (Negligible: V < 0.1, Weak: 0.1 ≤ V < 0.3, Moderate: 0.3 ≤ V < 0.5, Strong: V ≥ 0.5) [26].

Descriptive statistics were conducted in SPSS to give summary statistics and to analyse Likert scale questions. An eleven-point Likert scale questionnaire was used to determine survey participants’ knowledge, opinion, and experience of OFES, where 0 indicated zero knowledge, very low opinion, or very bad experience, and 10 indicated extensive knowledge, very good opinion, or very good experience of the procedure according to the question posed. 

Ordinal logistic regression was performed after dependent (outcome) and independent (predictor) variables were chosen for each factor that could influence farmers’ decisions concerning the management of acutely injured cattle and the use of OFES. Preliminary analysis was performed before regression analysis to assess whether the level of knowledge (dependent variable) displayed by farmers was influenced by the medium (independent variables) they used to obtain knowledge. It was also used to analyse if the matters discussed with other farmers and PVPs (independent variable) influenced farmers’ level of knowledge (dependent variable). Regression analysis was used again to assess if farmers’ opinion or experience of OFES (dependent variable) was influenced by what they considered as the positive and negative aspects of OFES (independent variable). The coefficient estimates B (the expected change in log odds), which describes the relationship between an independent variable and a response, was also calculated using ordinal logistic regression. A positive coefficient indicates that, as the value of the independent variable increases, the mean of the dependent variable also tends to increase and vice versa. Wald tests (Wald χ^2^) were used to determine if certain independent variables were significant. A *p*-value < 0.05 was deemed statistically significant for the purpose of the study.

The survey questionnaire is included in the Appendix A.

## 3. Results

In total, 107 survey responses were submitted. Thirteen respondents were excluded: nine respondents categorised their farming activity as ‘other’ and a further four did not categorise their farming activity. The analysis was based on 94 respondents, consisting of 49 dairy and 45 beef farmers. Not all respondents answered all questions.

### 3.1. Participant Demographics

Table 1 illustrates participant demographics. The main farming activity was dairy farming (52.1%, *n* = 49), followed by beef farming (47.9%, *n* = 45). The mean herd size was 170.6 (*SD* = 161.7) and dairy farms were, on average, 39% larger than beef farms (*M =* 209.2, *SD =* 168.8 dairy farmers, *M =* 127.6, *SD =* 143.1 beef farmers). The largest number of farms was located in Leinster (38.3%, *n* = 36) and Munster (31.9%, *n* = 30), with less representation from farmers in Connaught (16.0%, *n* = 15) and Ulster 9.6% (*n* = 9).

There was a significant correlation between age and gender (β = −0.2, *p* = 0.007, B = 1.4), indicating that female respondents were younger. There was no significant difference between the ages of dairy and beef farmers (*M =* 47.5, *SD =* 12.8 dairy farmers, *M =* 46.0, *SD =* 13.5 beef farmers).

A similar proportion of beef and dairy farmers had attained third level education (69.3% *n* = 34 dairy farmers, 64.4%, *n* = 29 beef farmers). Participation in the Knowledge Transfer Programmes operated by DAFM [28] was higher among dairy farmers than among beef farmers (71.4%, *n* = 35 dairy farmers, 43.2%, *n* = 19 beef farmers).

### 3.2. Capacity

In relation to where farmers obtained knowledge about the management of acutely injured cattle, their PVP was the most common source (43.5%, *n* = 10 dairy farmers, 40.0%, *n* = 24 beef farmers), followed by guidelines and regulations for beef farmers (17.4%, *n* = 4 dairy farmers, 21.7%, *n* = 13 beef farmers). Farmers appeared to use their PVP to inform them about regulations and guidelines pertaining to the management of acutely injured cattle. They also used farming organisations to inform them of guidelines (the farmer’s PVP and regulations: *r_s_* = 0.511, *n* = 94, *p =* < 0.001; farmer’s PVP and guidelines: *r_s_* = 0.514, *n* = 94, *p ≤* 0.001; farming organisations and guidelines: *r_s_ =* 0.330, *n* = 94, *p =* 0.001). There were no other significant correlations in relation to where farmers obtained their knowledge.

Table 2 illustrates the incidence of acutely injured cattle and how they were managed on respondent’s farms. The majority of beef farmers had no acutely injured cattle in 2020 (58.1% *n* =25), whilst 25.6% (*n* =11) reported that they had one acutely injured animal. In contrast, 27.0% (*n* = 13) of dairy farmers had no acutely injured cattle and 25.0% (*n* =12) reported that they had one acutely injured animal in 2020. In relation to euthanasia, the majority were performed by the Animal By-Products Collection Service (69.0%, *n* = 20), followed by PVPs (31.0%, *n* = 9).

In relation to the length of time since they had an acutely injured animal, the majority stated that it was over 3 years (46.1%, *n*= 6 dairy farmers, 60.0%, *n* = 12 beef farmers). The majority of respondents also indicated that they had no acutely injured cattle that required OFES (69.2%, *n* = 9 dairy farmers, 89.9%, *n* = 16 beef farmers) or CS (75.0%, *n* = 9 dairy farmers, 94.4%, *n* = 17 beef farmers), while 35.7% (*n* = 5) of dairy farmers and 89.9% (*n* = 16) of beef farmers had no acutely injured cattle euthanized in 2020.

### 3.3. Willingness

Table 3 summarises the factors that influenced the farmer’s decision-making relating to the management of acutely injured cattle and their experience of OFES. Survey participants reported that the views of their PVP was the main influencer when making decisions in relation to the management of acutely injured animals (45.6%, *n* = 10 dairy farmers, 55.5%, *n* = 10 beef farmers). In relation to positive aspects, the majority of dairy and beef farmers who responded to this question considered that OFES supports animal welfare (72.5%, *n* = 21 dairy farmers, 76%, *n* = 19 beef farmers). Regression analysis determined that overall, the relationship between both the farmer’s opinion and experience of OFES, and the positive and negative aspects of OFES, was not statistically significant. Furthermore, farmers’ opinion of OFES was not influenced by where they obtained their knowledge on managing acutely injured cattle. Spearman’s rank order correlation determined that there was a moderate positive correlation between farmers’ opinion and experience of OFES (*r_s_* = 0.506, *n* = 26, *p* = 0.008), indicating that their opinion of OFES became more positive with increased experience of OFES. Dairy farmers had a more positive experience and opinion of OFES (*Mdn =* 5.5, *IQR* 5 experience, *Mdn =* 8, *IQR* 4, opinion) compared to beef farmers (*Mdn =* 3, *IQR* 6 experience, *Mdn =* 4, *IQR* 4, opinion).

### 3.4. Opportunity

Regarding the nature of the injury of the last three acutely injured cattle reported by farmers, leg fractures were the most common (46.2%, *n* = 12 dairy farmers, 64.7%, *n* = 22 beef farmers), followed by pelvic injuries (46.2%, *n* = 12 dairy farmers, 11.8%, *n* = 4 beef farmers).

The highest financial impact for farmers were euthanasia and OFES, while CS was reported to have the lowest impact (Table 4). Dairy farmers reported a higher financial impact than beef farmers for all four approaches to managing acutely injured cattle, but the most marked difference was with CS (*Mdn =* 5, *IQR* 5 dairy farmers, *Mdn =* 2, *IQR* 7 beef farmers). Beef farmers perceived CS as the most economical method of dealing with acutely injured cattle, followed by treatment.

Table 5 summarises the opportunities for farmers to avail from OFES and their recommendations to increase uptake of OFES. 

A total of 53.8% (n = 7) of dairy farmers and 52.9% (n = 9) of beef farmers were not aware of any abattoirs providing for OFES within 100 km of their farm.

## 4. Discussion

The aim of the research was to explore farmers’ attitudes towards the management of acutely injured cattle, underpinned by their knowledge, willingness, and the opportunities available to them to employ OFES. The study adopted a theoretical framework in order to better understand the barriers to improving the management of acutely injured cattle. The results from this interdisciplinary study, and a related survey on veterinary perspectives [6], are intended to provide an initial evidence-base for official inspectors and policymakers about the issues pertaining to on-farm management of acutely injured cattle, and support initiatives to mitigate against non-compliance with regulatory requirements.

In total, 107 farmers responded to the survey, although 13 did not meet the selection criteria and were removed from the study. The small sample size and demographic of respondents do not represent the diversity of dairy and beef production systems in Ireland and can be considered limitations of the study. The respondents represent 0.07% of farmers in Ireland [29]. Their median age was 46.2 years, almost 90% were male, and the median herd size was 110 cattle; thus, older farmers with small holdings were not represented. Two thirds (67.0%) of respondents had a third level education, with similar results for both beef and dairy farmers. Almost two thirds had participated in the Knowledge Transfer Programme operated by DAFM [28], with dairy farmers having a significantly higher participation. Results from other surveys indicate that education in general contributes to “higher thinking” by farmers, regarding the social and environmental outcomes of actions that involve their agri-businesses [30]. Online survey-based methodology may not be an optimal means of engaging farmers, and the use of a paper-based survey may have been a better way to reach a broader cohort of respondents; however, this was not possible due to the COVID-19 pandemic. Results from published data show that, during the COVID-19 pandemic, ‘survey fatigue’ may have been an issue, as many researchers switched to non-contact-based research methods [31]. Another factor that may have influenced the response rate is that farmers may have felt discouraged to participate as they lacked or were unable to retrieve information in relation to the survey questions [31]. Furthermore, the response rate to some questions was poor and therefore, cannot be considered representative. Obtaining answers in relation to controversial questions about the management of acutely injured cattle may be difficult, as farmers may be concerned that the information could be used against them by regulators, by the media, or animal welfare organisations, even though it was highlighted that the survey was anonymous [32].

The Animal Identification and Movement database of Department of Agriculture Food and the Marine (DAFM) in Ireland show that, between 2020 and September of 2021, 651 cattle underwent OFES at Local Authority regulated abattoirs and 11 at DAFM regulated abattoirs. However, there is a degree of under reporting of cattle processed as OFES in the official data [33]. Results from the survey further indicated that the number of acutely injured cattle on respondent farms in 2020 was low, although not all farmers responded to this section. Six cases of OFES were reported by respondents, four for dairy farmers and two for beef farmers. In relation to CS, four cases were reported by respondents, three for dairy farmers and one for beef farmers, although the condition of these animals and their fitness for transport could not be ascertained. The number of OFES and CS cattle was low compared to numbers reported by PVPs and OVs [6]. Veterinarians reported a total of 343 OFES and 377 CS cattle in 2020 [6]. The reason for this discrepancy between the numbers reported by farmers and veterinarians could be due to a number of factors and would require further analysis. Furthermore, there was a lower response rate by farmers to questions relating to the management of acutely injured cattle, which may have also influenced survey results, and no respondent provided feedback in this regard. Two thirds of respondents reported that acutely injured cattle were euthanised by the Animal By-Products Collection Service. Irish farmers do not have a culture of euthanising their own injured cattle [34].

PVPs were reported to have a major role in influencing farmers when making decisions in relation to managing acutely injured cattle by whatever means available, including OFES and CS. Veterinary advice, along with a well-established relationship between farmers and PVPs, where there is a high level of trust, can lead to better compliance and better animal welfare outcomes [35]. A range of studies have identified the PVP as the main source of information and support on animal health issues [36]. Studies have also identified that the services delivered by PVPs are highly relative to the production system, the disease or management issue, and the individual farm [36]. The management of acutely injured animals is a reactive service provided by PVPs, but it can become a preventative service where PVP advice may be sought by the farmer to identify the hazards and reduce the risks of acute injuries of cattle. The farmer and the PVP have to work together to solve the issues on farms [36].

In contrast, almost one quarter of respondents had no discussion with their PVP, and this may indicate that there is not a strong farmer–veterinarian relationship and a low level of trust, in some cases. The farmer–veterinarian relationship is both a potential barrier and facilitator [37,38]. Farms can be challenging environments for PVPs due to physical hazards and financial constraints of the client. It is important that they remain alert to socio-economic factors when determining a course of action in the management of acutely injured cattle [39].

According to farmers who participated in the survey, acutely injured cattle were not common in Ireland in 2020. Results indicate that cattle are more likely to suffer acute injuries on dairy farms rather than beef farms. Experts rate dairy cattle as being at a higher welfare risk than beef cattle [40]; in terms of acute injury, this may relate to a number of management and environmental factors, such as more frequent handling and larger herd sizes [40]. This was the most likely reason that the incidence of pelvic fractures was higher in dairy cattle than in beef cattle. Dairy farmers had a higher opinion and a better experience of OFES, and this was possibly due to them having used the procedure to a greater extent. Participation in the Knowledge Transfer Programmes operated by DAFM [28] was higher amongst dairy farmers, and research has shown the importance of knowledge in identifying sick or injured animals [22].

In relation to the use of OFES, the majority of farmers had a positive opinion of the procedure, indicating that it was good for animal welfare. Whilst the response rate to the question was small, farmers indicated that the negative aspects of OFES related to the non-availability of the service and the lack of economic benefit. Regarding the latter, the financial impact of having to euthanise an acutely injured animal was considered similar to OFES. The price charged by abattoir owners to process OFES cattle, along with farmers not being paid adequately for injured cattle, does not always make OFES a viable option and may not allow farmers to recover production costs [11]. The financial impact was more marked for dairy farmers than beef farmers, and CS was considered to have the least financial impact; this was especially true for beef farmers. Such financial implications may act as a disincentive for farmers to use OFES for managing acutely injured cattle and support the continued use of CS, even where cattle may be unfit for transport.

Over half of all farmers stated that they were not aware of any abattoir in the vicinity of their farm providing OFES. Currently, there are 12 abattoirs in the Republic of Ireland permitting OFES of cattle, of which 50% are in two counties, four in Cork and two in Mayo. While subsidising OFES may be difficult to operate, the funding of small abattoirs to provide the service may be beneficial in enhancing availability and may also help to make the procedure more economically viable for farmers [41]. Another option to improve the availability is the establishment of mobile abattoirs to provide the service of OFES to farmers. Livestock, reindeer, and poultry are slaughtered in mobile abattoirs in several countries, but the total volume of mobile slaughter in the EU is not known [42]. EU legislation allows mobile slaughter of all kinds of domestic animals under Council Regulation (EC) No 1099 [2]. A further initiative that could help to improve the uptake of OFES is better communication between PVPs and those that could avail from the service but are sceptical in relation to the positive benefits for animal welfare [6,43]. The use of communication strategies, such as Motivational Interviewing among PVPs, could promote the use of OFES in managing acutely injured cattle [43].

## 5. Conclusions

The survey results indicate that farmers were positive towards the use of OFES in relation to animal welfare and aware of the relevant regulations and guidelines. However, several factors, mainly a low number of acutely injured cattle, lack of availability of abattoirs providing for OFES, and a perception that OFES offers little economic benefit, may have impeded the procedure being used extensively for managing acutely injured cattle. Improving the availability of OFES and subsidising the cost would reduce the number of acutely injured cattle that have to undergo euthanasia, CS, or possible treatment. Further research is required to understand the socioeconomic context of the management of acutely injured cattle and interventions to support good practice in Ireland. In addition, to provide an accurate picture of the management of acutely injured animals in Ireland, research is also required to ascertain how many acutely injured cattle are being disposed of by animal by-product operators. At a policy level, electronic data capture of the cause of death of cattle could provide clarity on the requirement for OFES by identifying the number of acutely injured cattle that are euthanised, go for CS, or undergo unsuccessful treatment. Monthly survey data are collected by DAFM Regional Veterinary Laboratories on cattle deaths, but this focuses on cause of death by disease and not on acute injury, as acutely injured cattle would not require pathology to determine the cause of death [44].

## Figures and Tables

**Table 1 animals-13-00450-t001:** Demographics of Dairy and Beef Farmer Respondents *.

Variable	Category	% Total (*n*)	% Dairy Farmers (*n*)	% Beef Farmers (*n*)
Gender	Men	86.2 (81)	89.8 (44)	82.2 (37)
	Women	13.8 (13)	10.2 (5)	17.8 (8)
Highest level of education	Tertiary	67.0 (63)	69.4 (34)	64.4 (29)
	Other	33.0 (31)	30.6 (15)	35.6 (16)
Participation in Knowledge Transfer Programmes	Yes	57.4 (54)	71.4 (35)	43.2 (19)
	No	41.5 (39)	28.6 (14)	56.8 (25)

* Not all respondents answered all questions.

**Table 2 animals-13-00450-t002:** Management of Acutely Injured Cattle by Dairy and Beef Farmers in Ireland *.

Variable	Category	% Total (*n*)	% Dairy Farmers (*n*)	% Beef Farmers (*n*)
Person responsible for performing euthanasia	Animal by-products collector	69.0 (20)	57.1 (8)	80.0 (12)
	Private veterinary			
	practitioner	31.0 (9)	42.9 (6)	20.0 (3)

Number of months since last acutely injured cattle	<6	3.0 (1)	7.7 (1)	0
	7–12	6.1 (2)	7.7 (1)	5.0 (1)
	13–24	21.2 (7)	23.1 (3)	20.0 (4)
	25–36	15.2 (5)	15.4 (2)	15.0 (3)
	>36	54.5 (18)	46.1 (6)	60.0 (12)

Number of acutely injured cattle in 2020	0	41.7 (38)	27.0 (13)	58.1 (25)
	1	25.3 (23)	25.0 (12)	25.6 (11)
	>1	33.0 (30)	48.0 (23)	16.3 (7)

Number of acutely injured cattle that were OFES in 2020	0	80.6 (25)	69.2 (9)	89.9 (16)
	1	19.4 (6)	30.8 (4)	11.1 (2)
	>1	0	0	0

Number of acutely injured cattle transported for CS in 2020	0	86.7 (26)	75.0 (9)	94.4 (17)
	1	13.3 (4)	25.0 (3)	5.6 (1)
	>1	0	0	0

Number of acutely injured cattle that required euthanasia in 2020	0	65.6 (21)	35.7 (5)	89.9 (16)
	1	15.6 (5)	35.7 (5)	0
	>1	18.8 (6)	28.6 (4)	11.1 (2)

* Not all respondents answered all questions.

**Table 3 animals-13-00450-t003:** Characteristics of Dairy and Beef Farmers Regarding the Management of Acutely Injured Cattle and On-Farm Emergency Slaughter (OFES): Willingness/Influences *.

Variable		Category	% Total (*n*)	% Dairy Farmers (*n*)	% Beef Farmers (*n*)
What influences decision-making in relation to managing acutely injured cattle		PVP	50 (20)	45.6 (10)	55.5 (10)
		Abattoir Owners	12.5 (5)	13.6 (3)	11.1 (2)
		Other Farmers	12.5 (5)	13.6 (3)	11.1 (2)
		Family	12.5 (5)	9.1 (2)	16.7 (3)
		Farming Organisations	2.5 (1)	4.5 (1)	0
		Other	10.0 (4)	13.6 (3)	5.6 (1)

Positive aspects of OFES		Good welfare outcomes	74.1 (40)	72.5 (21)	76.0 (19)
		Economic benefit to farmers	14.8 (8)	17.2 (5)	12.0 (3)
		Other	11.6 (6)	10.3 (3)	12.0 (3)

Negative aspects of OFES		Poor welfare outcomes	19.2 (5)	23.1 (3)	15.4 (2)
		Non availability of OFES	19.2 (5)	23.1 (3)	15.4 (2)
		No economic benefit	15.4 (4)	15.3 (2)	15.4 (2)
		Regulatory difficulties	19.2 (5)	7.7 (1)	30.7 (4)
		No prompt relief for welfare	11.6 (3)	7.7 (1)	15.4 (2)
		None	15.4 (4)	23.1 (3)	7.7 (1)
		Other	0	0	0

* Not all respondents answered all questions.

**Table 4 animals-13-00450-t004:** Financial Impact for Farmers of On-Farm Emergency Slaughter (OFES), Euthanasia, Casualty Slaughter (CS), or Treatment *.

Variable	Category	Median (IQR) (*n*)		
		Total	Dairy Farmers	Beef Farmers
Financial Impact of OFES	0 = very low, 10 = very high	7.5 (4) (22)	7.0 (4) (12)	8.0 (7) (10)
Financial Impact of Euthanasia	0 = very low, 10 = very high	7.5 (4) (22)	7.0 (4) (12)	8.5 (6) (10)
Financial Impact of CS	0 = very low, 10 = very high	4.5 (6) (22)	5.0 (5) (12)	2.0 (7) (10)
Financial Impact of Treatment	0 = very low, 10 = very high	5.0 (7) (22)	5.0 (6) (12)	4.5 (8) (10)

* Not all respondents answered all questions.

**Table 5 animals-13-00450-t005:** Availability of On-Farm Emergency Slaughter (OFES) and Farmer Recommendations for Improvements to the OFES Procedure *.

Variable		Category	% Total (*n*)	% Dairy Farmers (*n*)	% Beef Farmers (*n*)
Number of slaughterhouses within 100 km radius that provide the service of OFES		0	53.3 (16)	53.8 (7)	52.9 (9)
		1	40.0 (12)	38.5 (5)	41.2 (7)
		Other	6.7 (2)	7.7 (1)	5.9 (1)

Recommended changes to OFES procedure		More availability of OFES	13.7 (3)	33.3 (3)	0
		More availability of training	4.5 (1)	0	7.7 (1)
		More availability of information	18.2 (4)	11.1 (1)	23.1 (3)
		Make it more economical	31.8 (7)	33.3 (3)	30.7 (4)
		Other	31.8 (7)	22.2 (2)	38.5 (5)

Additional information about management of acutely injured cattle		Make it more economical	33.3 (2)	25.0 (1)	50.0 (1)
		Other	66.7 (4)	75.0 (3)	50.0 (1)

* Not all respondents answered all questions.

## Data Availability

The data presented in this study are available on request from the corresponding author.

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
