# Peer review of "Farmers Views on the Implementation of On-Farm Emergency Slaughter for the Management of Acutely Injured Cattle in Ireland"

_animals, 2023, doi:10.3390/ani13030450_

Round 1

Reviewer 1 Report

The title needs revision seeing the objective. (Willingness for OFES use). 

Introduction:

Which slaughter conditions are necessary using OFES and when is human consumption legal.

L65 The reasons for death and practice OFES and CS or euthanasia are not mentioned.  

added comments:

1. the main question addressed by the research:

A survey under farmers and vets is used to be informed about perceptions for using OFES, CS or euthanasia of injured cattle.

2. It address a specific gap in the field: No transport or accepted transport under supervision of a vet is a main welfare topic in injured cattle.

3.It add to the subject area: We do not know so much about the use of OFES by and perception of farmers and vets.

4.specific improvements: Use of OFES can be promoted by animal welfare organisations and government.

5.The conclusions are sound.

6.the references are appropriate.

7.The tables and figures are correct presented.

Author Response

Many thanks for reviewing our manuscript. We have addressed all of your comments as follows:

1. The title needs revision seeing the objective. (Willingness for OFES use). 

Thank you for your comment. Title amended to:

Farmers Views on the Implementation of on Farm Emergency Slaughter for the Management of Acutely Injured Cattle in Ireland

2. Which slaughter conditions are necessary using OFES and when is human consumption legal.

Thank you for your comment. Text on L 47-55 amended to:

In Ireland, for OFES to occur, abattoir owners must allow carcasses from acutely injured cattle that have been slaughtered on-farm into their abattoirs for processing. Provided that cattle are deemed suitable for OFES, based on an ante-mortem examination by a Private Veterinary Practitioner (PVP), slaughter is performed, typically on farm, by either a PVP, or a competent slaughter person as required by Regulation (EC) No 1099/2009 [2]. All parts of the carcass must be transported to the abattoir where a post-mortem examination is performed by an Official Veterinarian. If the meat from an OFES animal is passed as fit for human consumption it may be placed on the market. 

3. L65 The reasons for death and practice OFES and CS or euthanasia are not mentioned.  

Thank you for your comment, text amended on L74 to:

There is no policy of recording cause of death of cattle in Ireland.

Reviewer 2 Report

Lines 45-47: since this is a direct quote from somewhere, the source should be cited

Line 53: there is a closed quotation mark but not corresponding opening of the quote

Tables 2, 3, 5 have formatting issues

Author Response

Many thanks for reviewing our manuscript. We have addressed all of your comments as follows:

1. Lines 45-47: since this is a direct quote from somewhere, the source should be cited

Thank you for your comment,  source cited on L47;

  1. Regulation (EC) No 853/2004 of the European Parliament and of the Council of 29 April 2004 laying down specific hygiene rules for food of animal origin. 2004; Available online at: https://eur-lex.europa.eu/legal-content/EN/ALL/?uri=CELEX%3A32004R0853(accessed: 9 January, 2023).

2. Line 53: there is a closed quotation mark but not corresponding opening of the quote

Thank you for your comment, text amended on L60 - 61:

CS is “the slaughter at an abattoir of injured cattle that have been deemed fit for human consumption and fit for transport under veterinary certification” [5].

3. Tables 2, 3, 5 have formatting issues

Thank you for your comment. Tables 2, 3, and 5 have been amended to address formatting issues.

Reviewer 3 Report

Well written manuscript. Highlights some interesting international differences in the treatment of cattle.  

Tables 2, 3 and to some degree Table 5 have a poor matching of columns.  For example in Table 2: the Variable Number of Acutely Injured Cattle, the values in Category and % of Beef Farmers are not in line.

Author Response

Thank you for reviewing our manuscript and your comment:

Tables 2, 3 and to some degree Table 5 have a poor matching of columns.  For example in Table 2: the Variable Number of Acutely Injured Cattle, the values in Category and % of Beef Farmers are not in line.

Tables 2, 3, and 5 have been amended to address formatting issues. We will check with the Animals editorial team, in case of further formatting problems with the Tables (e.g.  if the formatting is altered when saved as a pdf on the Animals system. 

Reviewer 4 Report

The presentation is clear. There is some misalignment in the tables as printed (presumably it was OK in the manuscript)

I suggest you make it clear in the introduction that meat from OFES can enter the food chain subject to safeguards. Table 4 indicates that farmers expressed no difference between the economic costs of OFES and euthanasia.  In discussion you should consider in more detail why this should be so.

Author Response

Thank you for reviewing our manuscript. We have addressed all of your comments as follows:

1. The presentation is clear. There is some misalignment in the tables as printed (presumably it was OK in the manuscript)

Thank you for your comment. Tables 2, 3, 4 and 5 have been amended to address formatting issues.

2. I suggest you make it clear in the introduction that meat from OFES can enter the food chain subject to safeguards.

Thank you for your comment text amended on L 47-55:

In Ireland, for OFES to occur, abattoir owners must allow carcasses from acutely injured cattle that have been slaughtered on-farm into their abattoirs for processing. Provided that cattle are deemed suitable for OFES, based on an ante-mortem examination by a Private Veterinary Practitioner (PVP), slaughter is performed, typically on farm, by either a PVP, or a competent slaughter person as required by Regulation (EC) No 1099/2009 [2]. All parts of the carcass must be transported to the abattoir where a post-mortem examination is performed by an Official Veterinarian. If the meat from an OFES animal is passed as fit for human consumption it may be placed on the market. 

3. Table 4 indicates that farmers expressed no difference between the economic costs of OFES and euthanasia.  In discussion you should consider in more detail why this should be so.

Thank you for your comment. Text added to the Discussion L413 - 415:

The price charged by abattoir owners to process OFES cattle along with farmers not being paid adequately for injured cattle does not always make OFES a viable option and may not allow farmers to recover production costs [11];